# Mineral Texture Identification Using Local Binary Patterns Equipped with a Classification and Recognition Updating System (CARUS)

**Saeed Aligholi** [1,*] **, Reza Khajavi** [2] **, Manoj Khandelwal** [1] **and Danial Jahed Armaghani** [3]

1   Institute of Innovation, Science and Sustainability, Federation University Australia, Ballarat, VIC 3350, Australia
2   Earthquake Research Center, Ferdowsi University of Mashhad, Mashhad 9177948974, Iran
3   Department of Urban Planning, Engineering Networks and Systems, Institute of Architecture and Construction, South Ural State University, 76 Lenin Prospect, 454080 Chelyabinsk, Russia
*   Correspondence: s.aligholi@federation.edu.au

**Abstract:** In this paper, a rotation-invariant local binary pattern operator equipped with a local contrast measure (riLBPc) is employed to characterize the type of mineral twinning by inspecting the texture properties of crystals. The proposed method uses photomicrographs of minerals and produces LBP histograms, which might be compared with those included in a predefined database using the Kullback–Leibler divergence-based metric. The paper proposes a new LBP-based scheme for concurrent classification and recognition tasks, followed by a novel online updating routine to enhance the locally developed mineral LBP database. The discriminatory power of the proposed Classification and Recognition Updating System (CARUS) for texture identification scheme is verified for plagioclase, orthoclase, microcline, and quartz minerals with sensitivity (*TPR*) near 99.9%, 87%, 99.9%, and 96%, and accuracy (*ACC*) equal to about 99%, 97%, 99%, and 99%, respectively. According to the results, the introduced CARUS system is a promising approach that can be applied in a variety of different fields dealing with classification and feature recognition tasks.

**Keywords:** automated mineral identification; LBP; classification; texture feature

## 1. Introduction

There has been growing interest among researchers to develop well-performing automated schemes in different areas of science and engineering, including rock analysis, in recent years [1–8]. Automated mineral identification (AMI) as well as geometry characterization of rock constituents, as two prerequisites of any petrography scheme, are demanded in different fields of geosciences including rock mechanics, engineering geology, volcanology, mining, and underground construction [9–15]. Accordingly, some pattern recognition and image processing methods including textural and color analyses as well as frequency domain analysis are applied for mineral identification and rock classification [16–21]. The proposed schemes are mostly designed as color-based, rather than texture-based, and might fail to successfully identify twinned minerals. Such problems may also frustrate segmentation algorithms, which are increasingly used for the shape-and-size analysis of rock constituents. Texture analysis, as a chief ingredient of the automated mineral identification (AMI) task, is thus potentially favorable, or necessary, to develop any unified automated rock analysis package [17]. This study aims at the automation of identifying plagioclase, orthoclase, microcline, and quartz minerals through texture analysis. For the purpose of rock classification, identification of these minerals is an essential task, however, they might rarely be distinguished from each other.

Texture analysis is useful and is essentially required for some image processing purposes. A human can easily distinguish different textures, but it is a complicated machine

learning task [22]. Texture analysis deals with the following problems: image classification according to textural features; image segmentation; designing textures; and shape analysis from texture contents [23]. Texture classification is employed in different scientific problems including medical image analysis [24], fabrics analysis [25], and remote sensing [26].

Different techniques have been employed for the texture description of images. Texture analysis procedures have been classified into the following categories: geometric, statistical, signal processing, and model-based approaches [23]. Different typical texture features are also introduced in the literature [27,28]. Pioneering methods for texture classification were based on the statistical quantification of texture information, ranging from co-occurrence matrices [27] and polarograms [29] to procedures such as Markov random fields [30], signal processing methods [31–33], and local binary pattern [34], which offer satisfactory results.

The local binary pattern (LBP) is a statistical-based texture analysis operator for local texture characterization. LBP-based texture feature extraction has become increasingly applied for texture analysis during the last decade. LBP features have achieved considerable success in many fields, e.g., face recognition, remote sensing, and several medical image analysis problems [35–45]. Different extensions of the method have been developed for further improvements in power discrimination [46], rotation invariance [47,48], and robust noise resistance [49]. The operator (riLBPc) has some unique characteristics including computational simplicity, tolerance to grayscale, high discriminatory power, and illumination variations [34].

Mineral texture identification has been the subject of several studies published in the past; in reality, the majority of researchers have focused their attention on rock texture analysis. Ross et al. [20] have partly considered some texture attributes for the task of mineral identification through genetic programming. They used the intensity component of hue–saturation–intensity (HSI) space to calculate standard parameters such as energy, contrast, homogeneity, and entropy by means of a gray-level co-occurrence matrix, calculated for each grain. Thompson et al. [21] trained their neural networks with the above four texture parameters to identify colorless minerals: quartz, K-feldspar, and plagioclase. While their designed net could distinguish quartz and plagioclase well from each other, its ability to identify K-feldspar deteriorated, which might be a consequence of minor alterations such as sericitization. Smith and Beermann [50] attempt to identify plagioclase from quartz by employing gray-level homogeneity recognition.

Although the texture is a significant characteristic of feldspars, there is no solid model that quantifies the complex nature of these image properties. Therefore, a texture analysis approach is developed that attempts to explain the textural features of some minerals. The main focus of this study is on the identification of textural features that yield the highest retrieval accuracy. Rotation-invariant local binary patterns (LBPs) equipped with a complementary contrast measure are employed, to develop a so-called CARUS software for the task of automatic mineral identification of mineral textures. The scheme is also provided by a mineral texture database, which is enhanced after each mineral texture identification task. The results are promising and the texture of the studied minerals is successfully identified by employing CARUS with very high accuracy.

The paper is presented as follows. In Section 2, distinctive texture features (twinning and undoluse extinction) of the minerals plagioclase, quartz, and alkali feldspars (orthoclase and microcline) are introduced. Some different typical versions of LBP texture operators are briefly overviewed in Section 3. Section 4 introduces the two main kernels of the CARUS, the classification/recognition algorithm, as well as the proposed database updating system. In Section 5, several experiments to verify the performance of the CARUS algorithm for mineral pattern analysis and identification are described. Finally, the main conclusions drawn from this study are presented in Section 6.

## 2. Typical Textures in Minerals

Texture analysis is an inevitable task for the automated identification of the minerals located in the first order of the Michel-Lévy interference color chart (a chart that indi-

cates the interference color of minerals against the sample (thin section) thickness under cross-polarized light microscopy). Among these minerals, plagioclase, quartz, and alkali feldspars (orthoclase and microcline) are typically important for the development of any rock classification, e.g., [51,52]. These minerals usually have distinctive texture features (twinning and undoluse extinction). Twinning is a symmetrical intergrowth of two or more crystals of the same substance [53].

**Quartz (Qtz)**. In thin sections, quartz can be distinguished from twinned feldspars by its lack of twinning. Quartz in some igneous as well as metamorphic rocks indicates undulatory extinction because of strain. Therefore, patterns similar to fine lamellae may occur that are related to translation gliding [54]. Figure 1 (Q1,Q2) shows undoluse extinction in quartz grains.

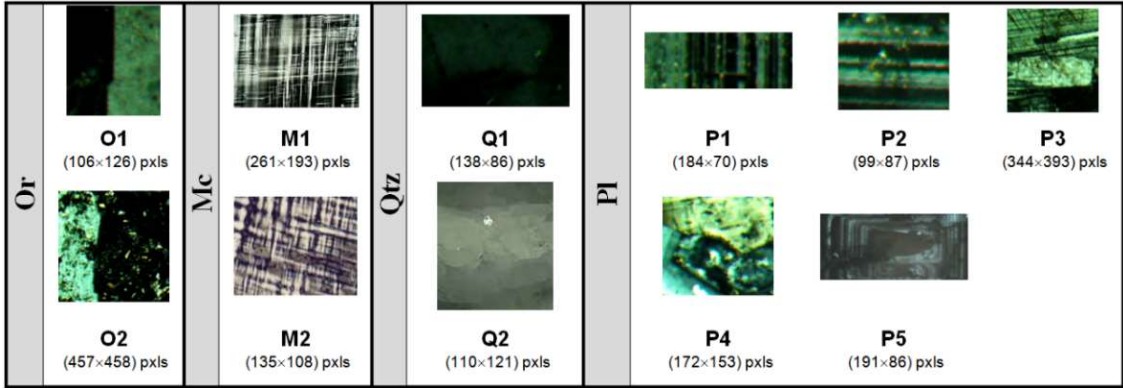

**Figure 1.** Examples of typical twinnings in real mineral images under XPL. (**O1,O2**) Orthoclase (Or); (**M1,M2**) microcline (Mc); (**Q1,Q2**) quartz (Qtz); and (**P1–P5**) plagioclase (Pl).

Twinning in feldspars varies according to the composition and the crystal system. Twinning in these minerals is the result of various mechanisms [54]: (a) it can occur during crystal growth, (b) it can be induced by deformation, and (c) it can be based on thermal transformation. There are different feldspathic twin laws. Feldspars commonly indicate three twin laws: normal, parallel, and complex. The feldspars best illustrate the twinning in the triclinic system. They are almost always twinned according to the albite law with the {010} twin plane and the pericline law with [010] as the twin axis. In the monoclinic system, twinning on {100} and {001} is most common. Carlsbad twinning was observed in the thin section as a pair of individual crystals, separated by a single line. Carlsbad twins may be either the interpenetrant or contact type. Carlsbad twinning is seen in monoclinic (such as sanidine and orthoclase) and triclinic feldspars (plagioclase and microcline).

**Orthoclase (Or)** can be distinguished from plagioclase feldspars by the absence of albite twinning and the frequent occurrence of simple Carlsbad twins. Orthoclase commonly shows simple twins. Figure 1 (O1,O2) shows simple twins in orthoclase grains.

**Microcline (Mc)** is characterized by a combination of albite and pericline twinning (tartan pattern) which is different from that found in albite. The combination produces a distinctive grid pattern and is particularly common in microclines because microclines are often formed by transformation. Sometimes, tartan twins are observed in plagioclase. However, in contrast to microcline, twin planes are well defined in plagioclase. Figure 1 (M1,M2) shows tartan twinning in microcline grains.

**Plagioclase (Pl)**. Multiple twinning is a distinguishing feature of all plagioclase feldspars. Carlsbad–albite, Carlsbad or simple, and pericline or repeated are also other typical twinning laws in Pl. Some plagioclase feldspars indicate zoning as a consequence of interior-to-outer variations in the composition of the crystal. Figure 1 shows multiple twins (P1–P3) and zoning (P4–P5) in plagioclase grains.

A modification, especially for the Or and Pl cases, may change typical texture features. Or usually changes to clay minerals; these minerals occur as discrete particles on the feldspar crystal. The amount and magnitude of the clay particles increase with alteration;

thus, they are usually termed sericite. Late-stage hydrothermal activity during solidification of the rock mass and chemical weathering is the main cause of Pl alteration. Zoisite minerals are also formed by feldspars during late-stage hydrothermal activity by a process called saussuritization [55]. In Section 5.1, changes in the LBP histogram of the studied minerals due to alteration will be discussed.

### 3. Brief Review of LBP

The grayscale local binary pattern (LBP) was introduced by Ojala et al. [56] and modified later to account for the rotation invariance with uniform patterns [57] to characterize the spatial structure. For any central pixel $(i,j)$ of the image with $n \times m$ pixels, the following LBP number is obtained through a simple comparison of the central pixel value with those of its $P$ neighboring pixels, located at a radius $R$ from it [47]:

$$LBP_{P,R}(i,j) = \sum_{p=0}^{P-1} H(g_p - g_c)2^p. \tag{1}$$

$H(x)$ is the discrete Heaviside step function, and $g_p$ and $g_c$ represent neighboring and central pixel grayscale values, respectively. In this study, $P$ is examined for the typical values of 8 and 16, with 1 and 2 for the radius value R; thus, a total of $2^8$ = 256 and $2^{16}$ = 65,536 different labels might be obtained depending on the gray levels of the center and the pixels in its neighborhood.

As proposed by the literature, the texture might be distinguished by both texture patterns and the strength of the patterns (contrast). Contrast is regarded as an important texture property for human vision, which entails useful grayscale-dependent information for the task of texture classification. However, the previous LBP operators totally ignore the magnitude of contrasts. A contrast measure (C) might be considered for each pixel by subtracting the average of the gray levels below the center pixel from those above (or equal to) it. For equal values of thresholded neighbors, the value of contrast is set to zero. It is proposed that *LBP* values divided by corresponding contrast (or variance) values will give better results as presented by Ojala et al. [57] and Pietikäinen et al. [58]:

$$LBPC_{P,R}(i,j) = LBP_{P,R}(i,j)/C(i,j). \tag{2}$$

The obtained *LBP* image of the sample mineral is represented as the following histogram:

$$h = \sum_{i=1}^{n} \sum_{j=1}^{m} f(LBP_{P,R}(i,j),k), \tag{3}$$

with the integer, $k \in [0,K]$, where $K$ is the maximum *LBP* value (e.g., $K = 17$ for $P = 16$), and the function $f$ is defined as:

$$f(x,y) = \begin{cases} 1 & x = y \\ 0 & otherwise \end{cases}. \tag{4}$$

The obtained histogram $h$ can easily be compared with a model histogram $h_0$ for possible similarity, using the following Kullback–Leibler divergence-based metric to evaluate their goodness-of-fit (*gof*) [28]:

$$gof_{h,h_0} = \sum_{i=1}^{N} h(i)logh(i) + \sum_{i=1}^{N} h_0(i)logh_0(i) - \sum_{i=1}^{N}(h(i) + h_0(i))(logh(i) + logh_0(i)) + 2log2. \tag{5}$$

In this study, uniform *LBP* operator (*U2*), as introduced by Mäenpää et al. [59], is used, in which the number of bitwise 0/1 transitions $U \leq 2$, where:

$$U = |H(g_{P-1} - g_c) - H(g_0 - g_c)| + \sum_{p=1}^{P-1} |H(g_P - g_c) - H(g_{p-1} - g_c)|. \tag{6}$$

In the uniform *LBP* operator, different output labels are assigned for uniform patterns, and a single label is assigned for the rest of the non-uniform patterns. This will return the output label number for patterns of $P$ bits as $P(P - 1) + 3$; e.g., for $P = 8$, the uniform *LBP* will produce 59 labels.

Since the rotation of any sample mineral image causes the *LBPs* to be shifted to different locations, the following rotationally invariant uniform *LBP* operator, rather than the one in Equation (1), might be employed:

$$LBP_{P,R}^{ri,\,U2}(i,j) = \begin{cases} \sum_{p=0}^{P-1} H(g_p - g_c), & U \leq 2 \\ P+1, & otherwise \end{cases} \tag{7}$$

## 4. CARUS Algorithm for MI

Precise rock characterization and classification require correct identification of quartz and feldspars, that might be recognized from each other based on typical texture feature observation through optical microscopy of rock thin sections in cross-polarized light (XPL); so, texture analysis is an inevitable task for automation of rock photomicrograph analysis. As noted in Section 2, mineral texture patterns are sometimes polluted by chemical (or instrumental) factors and may thus show atypical texture features; true classification/recognition of such minerals might fail unless a reasonable degree of algorithm flexibility is considered. The proposed algorithm is designed with this problem in mind, with an adaptive sense to incorporate algorithm flexibility and robustness. Details of the proposed CARUS algorithm are presented as follows.

### 4.1. Classification and Recognition

In manual MI through optical microscopy, the experienced operator attempts to identify texture features of minerals by examining the nature of the changes observed in the appearance of the unknown mineral, as the sample (rock/mineral thin section) is rotated with respect to the polarizers under XPL illumination. In automated MI for modeling rotation of a sample with respect to the polarizers, some photomicrographs of the sample in different rotations can be taken, e.g., [16,20]. This study aims at the automation of such mineral texture classification/recognition through texture analysis with the use of an appropriate LBP operator. For this purpose, the texture-based mineral identification (TMI) Algorithm 1 and Figure 2 is designated. The procedure of the algorithm is as follows:

(a) For an unknown mineral, an appropriate XPL image (usually the one with the maximum standard deviation (maxSD_Img)) from a set of rotated images is selected. This image is commonly expected to show typical textural features of a mineral.
(b) The histogram (*h*) of its LBP image (LBP_Img) with *nb* bins is obtained (Equation (7)). Usually, *nb* is considered to be 100 × (P + 1), where P is the number of LBP samples as introduced in Section 3.
(c) The *h* histogram of the LBP_Img is compared with those included in the ***h-db*** (a database of LBP image histograms of *nt* (*i*) typical samples for any mineral class i ≤ *nc*) through a distance metric gof, as introduced in Equation (5). Steps 1 through 5 of the TMI algorithm will provide a ***GOF*** matrix which incorporates the *gof* values corresponding to the *h* of the unknown mineral and those included in the ***h-db***.
(d) In steps 6 through 8, a mean function might be used, as shown in Figure 3, to develop a ***gof*** vector which specifies the *gof* values of the unknown mineral with any mineral classes included in ***h_db***. The averaging scheme, as will be verified in Section 5, does have enough accuracy, increases robustness (in comparison with methods such as k-nearest neighbor (kNN)), and reduces the computational cost.
(e) In steps 9 and 10, the minimum entry of ***gof*** is found, and its corresponding mineral class *c* is introduced as the one to which the unknown mineral belongs.

**Algorithm 1.** Pseudo code for texture-based mineral identification algorithm (TMI).

**Input:** unknown mineral XPL image with standard deviation (maxSD_Img)
**Output:** mineral class of the unknown mineral (c)
**Initialization: find** $h \equiv \{h_i\}^{nb}_{i=1}$ from LBP image of unknown mineral image

```
 1 :   for i ← 1 : nc do
 2 :     for j ← 1 : nt(i) do
 3 :       GOF(i, j) ← gof[h, h_j^i ε h_db]
 4 :     end for
 5 :   end for
 6 :   for i ← 1 : nc do
 7 :     gof(i) ← mean[{GOF(i, j)}^{nt(i)}_{j=1}]
 8 :   end for
 9 :   gof ← min[{gof(i)}^{nc}_{i=1}] = gof(c)
10 :  return⟨mineral class⟩c
```

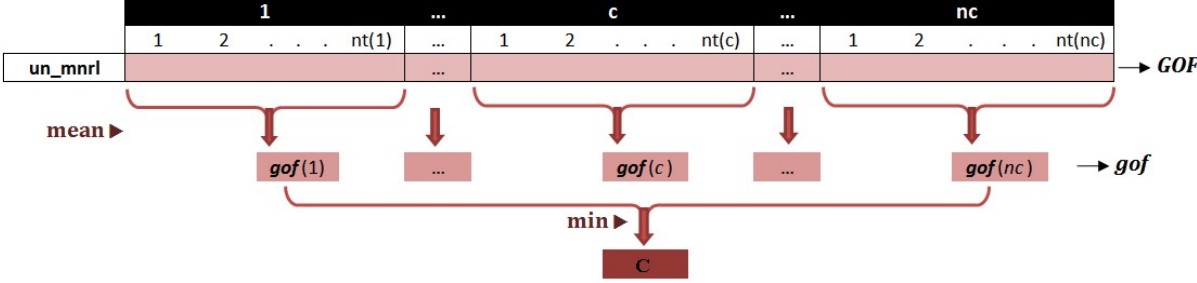

**Figure 2.** Schematic view of TMI algorithm.

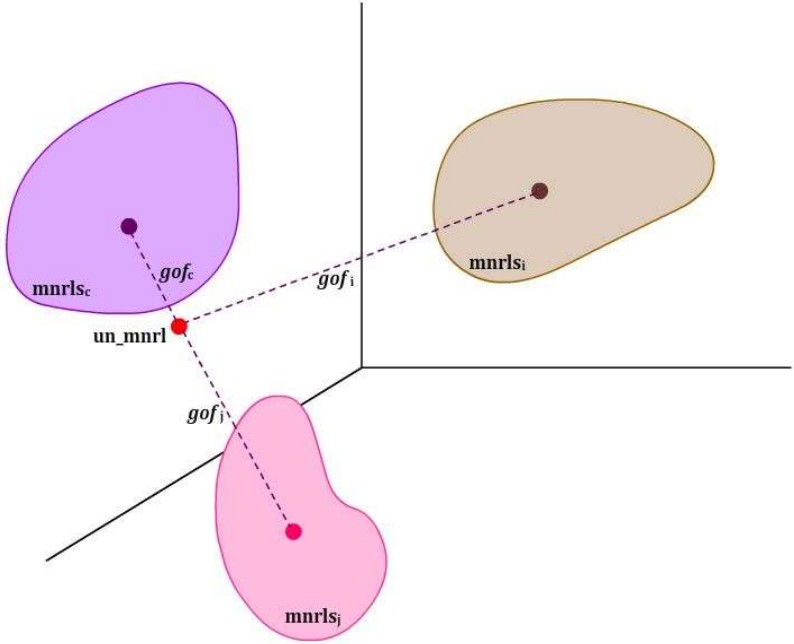

**Figure 3.** Schematic view of mean *gof*s between the unknown histogram and those of mineral classes (points represent histograms).

### 4.2. Updating System

An important prerequisite of the above texture-based mineral identification scheme is a database of LBP histograms obtained for some typical samples of the mineral classes, ***h_db***. In contrast to some pattern recognition applications such as face recognition, for which

several standard databases have been developed, no unified standard mineral database has been developed yet in the mineralogy community, and research studies for MI are inevitably based upon local databases. This means that any research group starts with a limited number of available mineral instances, which can be developed as more instances are inspected.

In this study, a second algorithm for database updating (HDU), illustrated in Algorithm 2 and Figure 4, is proposed to successively update an initial *h_db* after each run of the TMI scheme. The *h_db* updating algorithm is as follows:

(a)    An initial database of LBP histograms, *h_db*, obtained from an initial collection of some typical mineral samples, is established. In developing such a dataset, mineral images must be selected such that they can well represent different types of structural and non-structural conditions. While structural conditions (S factors) pertain to both the genesis of the mineral (S-I factors) and the environmental conditions such as temperature, pressure, and hydrothermal solution (S-II factors) that cause different degrees of alteration, the non-structural ones (NS factors) are mostly related to laboratory and equipment conditions (e.g., microscopes and digital cameras with different qualities, lighting changes, different magnification in microscopy, different orientations, etc.). It will be shown that a poor initial database may deteriorate the performance of the developed database.

(b)    A *GOFc* matrix, incorporating all gof values obtained for the sample minerals of the *h_db* belonging to the *c*th mineral class, is established (steps 1 through 5).

(c)    Mean gof values for each sample of the mineral class *c* are calculated, which gives the vector *gofc* (steps 6 through 8); a *gofc_s* vector is obtained by sorting *gofc* in descending order (step 9).

(d)    In steps 10 through 14, the HDU algorithm decides whether to add the LBP histogram of the identified mineral sample (in the previous TMI run) to *h_db* or not. For this purpose, the two following conditions are investigated:

(1)    gof value of the identified sample must be less than *gofc_s*(1) which is the maximum entry of the *gofc* vector. The condition is schematically illustrated in Figure 5a. LBP histograms of mineral instances in class *c*, **mnrls$_c$**, are within an acceptance circle $\mathbf{C}^1_{max}$ with the radius of *gofc_s*(1) (the center of the circle is a virtual point and is depicted for geometric interpretation). As the first condition, the identified mineral, **un_mnrl**, must be located within the acceptance circle of **mnrls$_c$**.

(2)    *gofc_s*(1) is checked for being less than all gof values of the *gof* vector (other than *gof* =*gof*(*c*)); it is reminded that *gof* is obtained in steps 6–8 of the TMI algorithm (refer to Algorithm 1 (and Figure 3)). The geometric interpretation of this criterion is illustrated in Figure 5a, where the mean virtual points of all mineral classes (except c) are located out of the dashed circle $\mathbf{C}'_{max}$ (a circle with the center point of un-mnrl, and the same radius as $\mathbf{C}^1_{max}$). If this condition is satisfied, an LBP histogram of the identified sample *h* is added to *h_db*. Otherwise, the next entry of *gofc_s*, i.e., the next maximum value of *gofc*, is examined for this criterion. This loop is continued until both conditions are satisfied. Figure 5b illustrates a situation where the second criterion is not satisfied (i.e., both virtual centers of **mnrls$_c$** and **mnrls$_j$** classes are within the light dashed circle), but by selecting the *k*th maximum entry of *gofc_s*, a circle $\mathbf{C}'_{max}$ (with the same radius as $\mathbf{C}^k_{max}$) is introduced for which centers of **mnrls$_c$** and **mnrls$_j$** are, respectively, within and out of it. There might be cases for which such reduction in the acceptance circle may fail the first condition (i.e., the identified mineral, **un_mnrl**, located out of the acceptance circle ($\mathbf{C}'_{max}$) of **mnrls$_c$**); thus, the mineral will not be added to the database.

---

**Algorithm 2.** Pseudo code for *h-db* update algorithm (HDU).

---

**Input:** *c*, *gof*, *h_db*
**Output:** update *h_db*

1 :   **for** $i \leftarrow 1 : nt(c)$ **do**
2 :    **for** $j \leftarrow 1 : nt(c)$ **do**
3 :      $GOFc(i,j) \leftarrow \text{gof}\left[h_i^c \, \epsilon \, h\_db, h_j^c \, \epsilon \, h\_db\right]$
4 :    **end for**
5 :   **end for**
6 :   **for** $i \leftarrow 1 : nt(c)$ **do**
7 :    $gofc(\text{i}) \leftarrow \text{mean}\left[\{GOFc(i,j)\}_{j=1}^{nt(i)}\right]$
8 :   **end for**
9 :   $gofc\_s \leftarrow \text{sort}[gofc, \text{ descending}]$
10 :  **for** $i \leftarrow 1 : nt(c)$ **do**
11 :   **if**$\left(gof = gof(c) < gofc\_s(i) < \{gof(i)\}_{i=1, i\neq c}^{nc}\right)$ **then**
12 :    add *h* to *h_db*
13 :   **exit**
10 :  **end for**

---

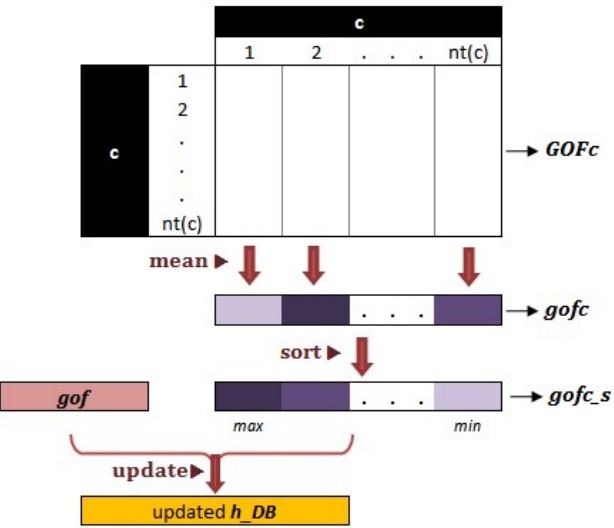

**Figure 4.** Schematic view of HDU algorithm.

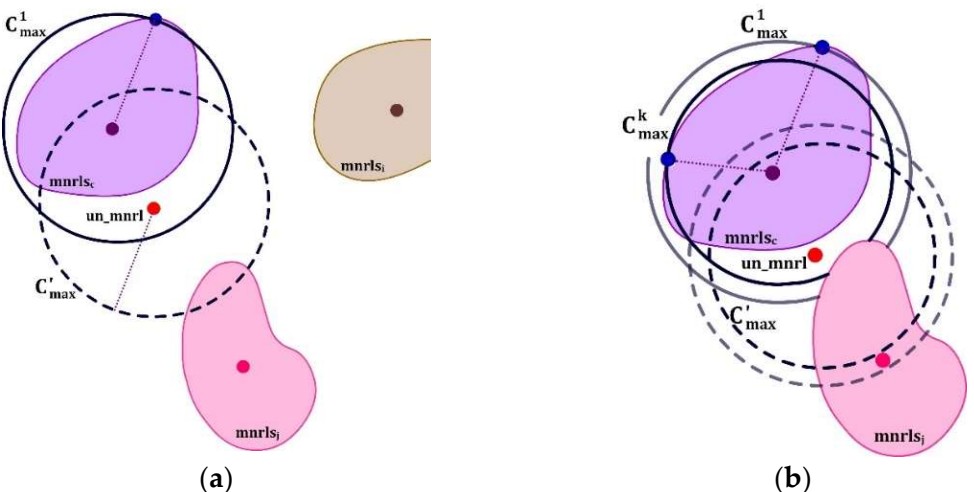

<p style="text-align:center">(**a**)         (**b**)</p>

**Figure 5.** Geometric interpretation of updating step of the CARUS: (**a**) un_mnrl is added to the database, and (**b**) un_mnrl is added to the database after reducing the acceptance circle.

## 5. Experiments and Results

Different experiments are carried out to indicate the performance of different aspects of the CARUS algorithm for mineral pattern analysis and identification. In Section 5.1, different local binary patterns are examined and investigated, through a simple example, to give an essence of their performance in dealing with mineral images. It will be shown that the rotation-invariant LBP equipped with contrast (riLBPc) is the most well-performing compared to the others, and is employed as the representative LBP variant for the task of mineral pattern identification. In Section 5.2, the proposed TMI scheme is tested and compared with the routine of Haralick's gray-level co-occurrence matrices (GLCMs). The performance of the CARUS algorithm (with the updating part included) will then be verified in Section 5.3. In Section 5.4, the CARUS classification scheme and the well-known k-nearest neighbor method are compared in terms of how well they work.

### 5.1. Comparison of LBPs for TMI

In this section, the accuracy and performance of four different LBP schemes: basic LBP (bLBP), uniform LBP (uLBP), rotation-invariant LBP (riLBP), and rotation-invariant LBP equipped with contrast (riLBPc), are examined for the task of texture identification of the four minerals under consideration, i.e., orthoclase, plagioclase, microcline, and quartz. For comparing the four schemes, 11 sample minerals from Figure 1 are employed. Figure 6 illustrates the LBP (8,1) images and their corresponding histograms of each sample mineral, obtained by the four schemes. In Figure 7, the *gof* values obtained for any pair of the sample minerals in Figure 1 are reported. For the sake of simplicity, the limit value of 0.2 is considered for the discrimination of the minerals; i.e., *gof* > 0.2 implies dissimilarity between the twinning patterns. It is, however, mentioned that using 0.2 as a fixed threshold for feature discrimination is not robust, and just gives an essence of the performance of different methods. In Figure 7, the *gof* values larger than 0.2 are highlighted for better comparison.

The bLBP section in Figure 7a shows the results of the conventional basic local binary patterns, as proposed by Ojala et al. [56]. While *gof* values for most similar minerals are less than 0.2, dissimilar minerals are not well discriminated. The scheme, as is expected, is not reliable and robust for the task of texture classification of minerals.

Using some uniform patterns rather than all patterns enhances recognition results for most applications. uLBP seems to be more statistically robust and stable (i.e., less prone to noise), while it requires fewer LBP labels [58]. Texture noise is an inevitable part of thin-section mineral images. Usually, due to the distributed (or even point-wise) regions of alteration in orthoclase or plagioclase, some non-uniform patterns might appear; such image contamination can be alleviated by the use of uBLP. However, if alteration, as a typical feature of Pl or Or, spreads over a large region of the sample, it should be considered as a texture feature of the mineral. High gof values that may appear for the samples of the same mineral could be associated with this. Microscope magnification, as well as the resolution of the camera, might intensify such errors. For example, if a thin section with point-wise alteration is observed with high magnification, the alteration spreads over the image, giving a different gof. The same situation appears when an improved camera resolution is applied. Thus, the alteration area is dependent on the magnification of the microscope, as well as the camera resolution. Though uLBP performs nearly as well in alleviating noise effects and dealing with parameters such as alteration, exsolution, fractures, and veins, it fails to effectively categorize minerals, mostly due to its sensitivity to orientation.

The abovementioned deficiencies of the previous two LBP schemes must partly be associated with their inability to deal with the orientation. Different orientations of the input image cause the LBPs to translate into different locations while rotating about their origin. Translation can be normalized by computing the histogram of LBP codes, and rotation can be normalized by using a simple rotation invariant scheme [58]. A comparison of the results of the riLBP method with those of the two previous ones in Figure 7 clearly shows that errors due to dissimilar texture orientations have been efficiently reduced. For example,

the gof values for P1 and P2 (which are well characterized for their nearly perpendicular orientations) are 0.25 and 0.22 in bLBP and uLBP schemes, respectively; however, the value has decreased to 0.09 by riLBP. From Figure 7, it can easily be verified that except for the pair of Pl–Mc, the gof values for dissimilar minerals have generally increased. It must be pointed out that both Pl and Mc are crystallized in a triclinic system, with albite and plagioclase twinnings; however, Pl usually exhibits one of such twinnings perfectly, while Mc, which is often formed by transformation from Or as a consequence of temperature decrease, shows both twinnings in a pinch and swell manner. Thus, the textural features of the two minerals are similar. This might easily be examined by comparing the LBP images of the two minerals illustrated in Figure 6. Thus, their LBP codes are similar, especially when normalized to 1.

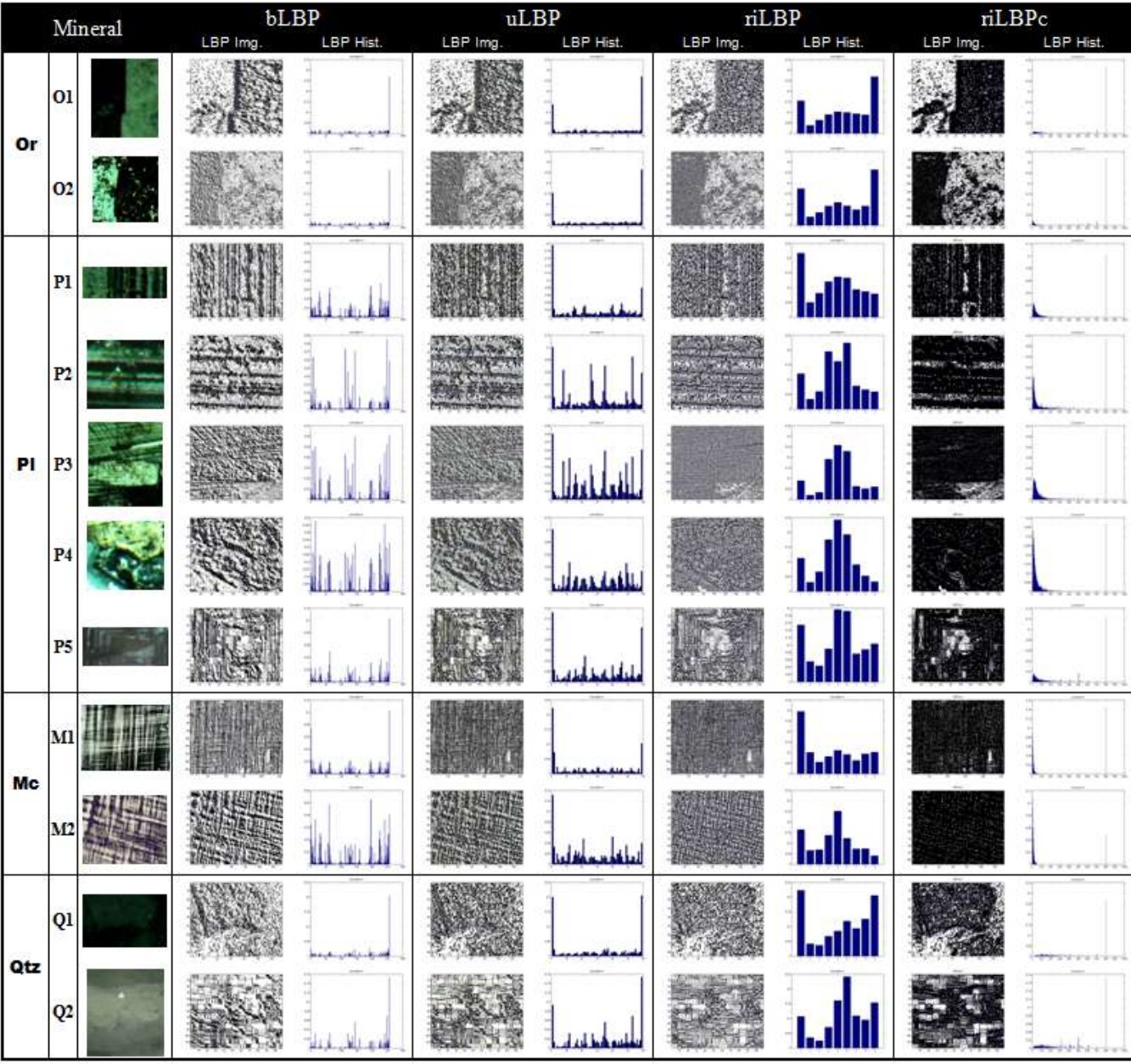

**Figure 6.** LBP (8,1) images and their corresponding histograms for sample minerals of Figure 1, obtained for different LBP schemes: basic LBP (bLBP), uniform LBP (uLBP), rotation-invariant LBP (riLBP), and rotation-invariant LBP equipped with contrast (riLBPc).

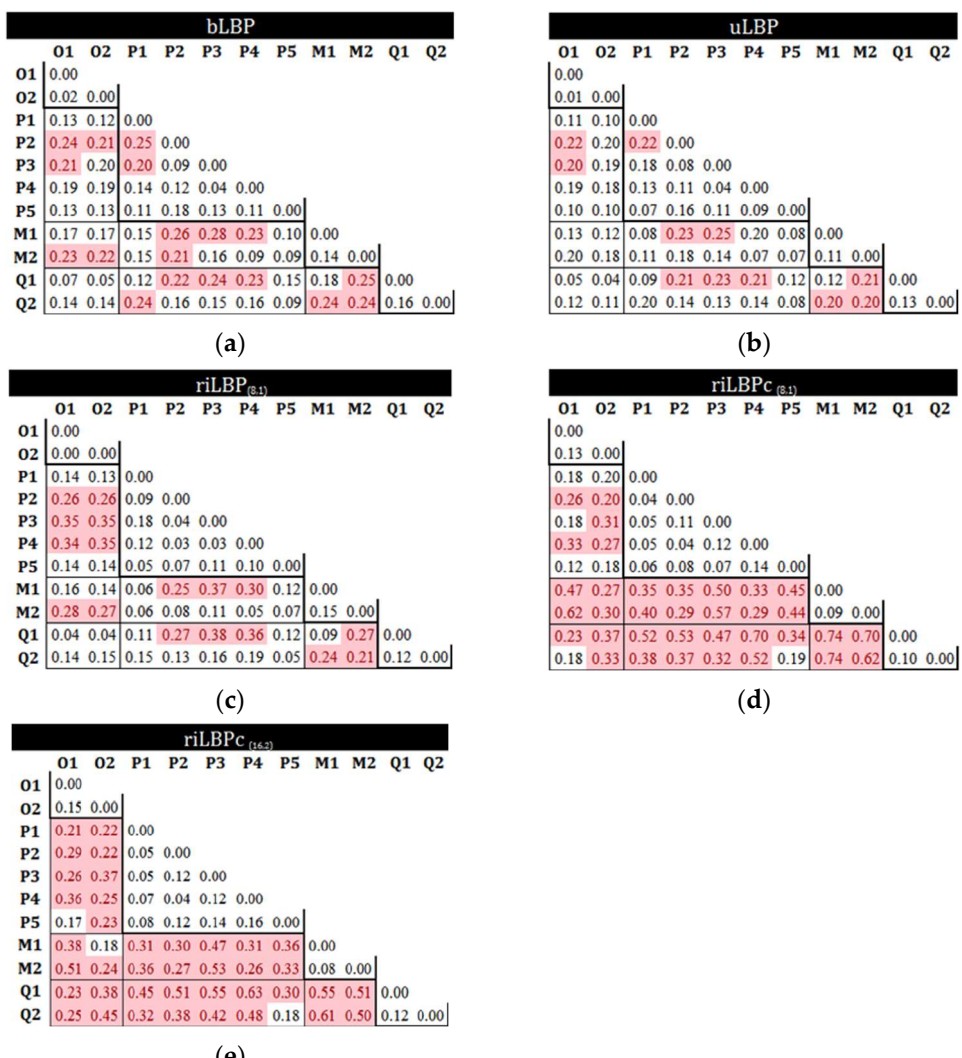

**Figure 7.** gofs obtained for any pair of the sample minerals of Figure 1, obtained for different LBP schemes: (**a**) bLBP, (**b**) uLBP, (**c**) riLBP, (**d**) riLBPc (8,1), and (**e**) riLBPc (16,2).

For better mineral texture classification of similar minerals such as Pl–Mc, the contrast feature is employed, which is usually more in Mc samples compared with the Pl ones, mostly due to their chemical composition and different birefringence. An alteration may also produce some different colors in Pl which helps in recognizing the mineral if contrast is used.

From Figure 7, it can be clearly seen that consideration of the contrast (using LBPc scheme according to Equation (2)) has efficiently improved the results. The *gof* values for the samples belonging to the same minerals have generally decreased, while those associated with dissimilar ones have mostly increased, especially for minerals such as Mc and Qtz. However, for some minerals such as Pl and Or which usually have the same contrast and alteration, the results have slightly deteriorated in some cases. O1 and O2 have small *gof*s before using contrast because O2 is affected by alteration and has changed to secondary minerals with high birefringence (light colors), which are characteristically in contrast with unaltered parts.

After that, the riLBPc version of the local binary pattern is selected as the texture descriptor of the minerals. Two options of (8,1) and (16,2) are examined for the P–R pair introduced in Equation (1). Figure 7 verifies that riLBPc (16,2) has well discriminated Or from Pl, and Or from Qtz, and performs much better than riLBPc (8,1). However, the scheme is a bit less efficient than riLBPc (8,1) in the discrimination of Or from Mc, since

the gof values given by riLBPc (8,1) are greater than those of riLBPc (16,2). As presented in Figure 7, the *gof* magnitudes obtained for mineral samples of the same class have not changed much. As a final conclusion of this section, the riLBPc (16,2) is employed for the task of texture recognition in the process of mineral identification.

### 5.2. Comparison of LBP with GLCM

Haralick et al. [27] proposed the gray-level co-occurrence matrix (GLCM) analysis based on the idea that texture data are contained in the spatial arrangement of gray level values. The GLCM is measuring how often various combinations of neighboring pixel values occur. GLCMs can be used to obtain statistical features that characterize the texture. To capture texture properties, we chose a subset of four features: energy, contrast, correlation, and homogeneity. In this study, level B is set to 8, following studies focused on texture analysis by means of GLCM [32].

The immediate neighboring pixels are considered along the four directions of 0°, 45°, 90°, and 135° [60], with the distance between the center and neighboring pixels, δ, set to 1 or 2. Therefore, for a considered δ, a GLCM composed of 16 descriptors (four statistical features for four different orientations) is obtained for characterizing mineral textures. A multi-scale feature pattern of dimension 32 (GLCM) is then defined by integrating the descriptors obtained from the two considered values of δ. Figure 8a shows the mutual distances of the feature vectors for the 11 sample templates of Figure 1. It is notable that amongst these 4 features, homogeneity shows the best performance. Figure 8b demonstrates the results of the *gof*s obtained from the riLBPc (16,2) scheme. A simple comparison between the two figures shows that the latter is more efficient for the task of mineral texture classification.

| Mineral | Mineral Temp. | Or O1 | Or O2 | Pl P1 | Pl P2 | Pl P3 | Pl P4 | Pl P5 | Mc Mc1 | Mc Mc2 | Qtz Q1 | Qtz Q2 |
|---|---|---|---|---|---|---|---|---|---|---|---|---|
| Or | O1 | 0.00 | 0.18 | 0.47 | 0.49 | 0.40 | 0.44 | 0.47 | 1.00 | 0.88 | 0.24 | 0.30 |
| | O2 | 0.18 | 0.00 | 0.34 | 0.34 | 0.29 | 0.27 | 0.33 | 0.82 | 0.70 | 0.17 | 0.22 |
| Pl | P1 | 0.47 | 0.34 | 0.00 | 0.06 | 0.09 | 0.10 | 0.05 | 0.53 | 0.41 | 0.39 | 0.22 |
| | P2 | 0.49 | 0.34 | 0.06 | 0.00 | 0.10 | 0.09 | 0.08 | 0.51 | 0.39 | 0.41 | 0.24 |
| | P3 | 0.40 | 0.29 | 0.09 | 0.10 | 0.00 | 0.07 | 0.13 | 0.60 | 0.48 | 0.39 | 0.18 |
| | P4 | 0.44 | 0.27 | 0.10 | 0.09 | 0.07 | 0.00 | 0.14 | 0.56 | 0.44 | 0.38 | 0.19 |
| | P5 | 0.47 | 0.33 | 0.05 | 0.08 | 0.13 | 0.14 | 0.00 | 0.53 | 0.41 | 0.38 | 0.22 |
| Mc | Mc1 | 1.00 | 0.82 | 0.53 | 0.51 | 0.60 | 0.56 | 0.53 | 0.00 | 0.21 | 0.91 | 0.75 |
| | Mc2 | 0.88 | 0.70 | 0.41 | 0.39 | 0.48 | 0.44 | 0.41 | 0.21 | 0.00 | 0.80 | 0.63 |
| Qtz | Q1 | 0.24 | 0.17 | 0.39 | 0.41 | 0.39 | 0.38 | 0.38 | 0.91 | 0.80 | 0.00 | 0.22 |
| | Q2 | 0.30 | 0.22 | 0.22 | 0.24 | 0.18 | 0.19 | 0.22 | 0.75 | 0.63 | 0.22 | 0.00 |

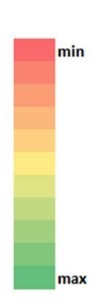

(a)

| Mineral | Mineral Temp. | Or O1 | Or O2 | Pl P1 | Pl P2 | Pl P3 | Pl P4 | Pl P5 | Mc Mc1 | Mc Mc2 | Qtz Q1 | Qtz Q2 |
|---|---|---|---|---|---|---|---|---|---|---|---|---|
| Or | O1 | 0.00 | 0.24 | 0.33 | 0.47 | 0.42 | 0.57 | 0.27 | 0.61 | 0.81 | 0.37 | 0.40 |
| | O2 | 0.24 | 0.00 | 0.35 | 0.34 | 0.60 | 0.40 | 0.37 | 0.29 | 0.38 | 0.61 | 0.72 |
| Pl | P1 | 0.33 | 0.35 | 0.00 | 0.08 | 0.08 | 0.11 | 0.12 | 0.49 | 0.57 | 0.72 | 0.51 |
| | P2 | 0.47 | 0.34 | 0.08 | 0.00 | 0.19 | 0.06 | 0.19 | 0.48 | 0.43 | 0.81 | 0.61 |
| | P3 | 0.42 | 0.60 | 0.08 | 0.19 | 0.00 | 0.20 | 0.23 | 0.76 | 0.84 | 0.87 | 0.67 |
| | P4 | 0.57 | 0.40 | 0.11 | 0.06 | 0.20 | 0.00 | 0.26 | 0.49 | 0.42 | 1.00 | 0.76 |
| | P5 | 0.27 | 0.37 | 0.12 | 0.19 | 0.23 | 0.26 | 0.00 | 0.58 | 0.52 | 0.48 | 0.29 |
| Mc | Mc1 | 0.61 | 0.29 | 0.49 | 0.48 | 0.76 | 0.49 | 0.58 | 0.00 | 0.12 | 0.88 | 0.97 |
| | Mc2 | 0.81 | 0.38 | 0.57 | 0.43 | 0.84 | 0.42 | 0.52 | 0.12 | 0.00 | 0.82 | 0.79 |
| Qtz | Q1 | 0.37 | 0.61 | 0.72 | 0.81 | 0.87 | 1.00 | 0.48 | 0.88 | 0.82 | 0.00 | 0.19 |
| | Q2 | 0.40 | 0.72 | 0.51 | 0.61 | 0.67 | 0.76 | 0.29 | 0.97 | 0.79 | 0.19 | 0.00 |

(b)

**Figure 8.** Results of: (**a**) combined Haralick multi-scale feature, and (**b**) rotation-invariant LBP (16,2) with a complementary contrast measure. (Numbers are normalized to 1 for the sake of comparison).

To compare the performance of different variants of LBP with the method of GLCM, a set of 200 RGB images of 50 microcline, 33 orthoclase, 57 plagioclase, and 60 quartz mineral samples are provided in the PNG format. The mineral dataset is split into 10 folds, each containing 20 sample images with roughly five images per class. The system uses each fold as an initial dataset, and the other nine folds (the remaining 180 images) as a test set. The

obtained classification results based on different texture operators and features are illustrated in Table 1. kNN is used as the classifier with Euclidean distance for all operator variants. The discriminatory power values reported are obtained as the mean magnitudes of results of the above 10 test sets. According to Table 1, both riLBPc (8,1) and (16,2) are clearly more efficient than the others. It is also interesting to note the effect of contrast, which has efficiently improved the discriminatory power in comparison with the riLBP operator.

**Table 1.** Discriminatory power results in different texture operators.

| Method | Discriminatory Power |
|---|---|
| bLBP | 0.61 |
| uLBP | 0.645 |
| riLBP$_{(8,1)}$ | 0.745 |
| riLBPc$_{(8,1)}$ | 0.82 |
| riLBPc$_{(16,2)}$ | 0.825 |
| GLCM$_1$ | 0.75 |
| GLCM$_2$ | 0.77 |

*5.3. CARUS Algorithm Validation*

In this section, the proposed algorithm is examined to verify its performance for the task of texture identification of the minerals based on database updating. For this purpose, the 10-fold dataset, introduced in the previous section, is used. Again, each fold is considered as the initial database and the other folds as test ones. The CARUS algorithm is run 10 times successively, which means that each image is included once in the initial database and there is no bias between the initial database and test samples.

For each unknown mineral, the TMI scheme is employed to determine the type of twinning class (*c*). After the mineral identification step, the algorithm is performed to check whether the identified mineral might be included in the database or not. Initial databases, wrongly added minerals (minerals that are falsely included in other classes), and the minerals which are not appended to the database are shown in Figure 9, while in Figure 10, convergence diagrams obtained by 10 successive runs of the above procedure for each fold are illustrated. The convergence diagrams for any mineral classes of Or, Pl, Mc, and Qtz, as well as the one for the whole **h_db**, are included. The three sets of convergence diagrams are associated with the **h_db** size and the two following values of discriminatory power including sensitivity or true positive rate (*TPR*) and accuracy (*ACC*):

$$TPR = \frac{TP}{TP + FN} \tag{8}$$

$$ACC = \frac{TP + TN}{TP + FN + FP + TN} \tag{9}$$

where *TP* is the number of correct predictions and *FN* is the number of incorrect predictions that an instance is positive, while *FP* is the number of correct predictions and *TN* is the number of incorrect predictions that an instance is negative. As will be shown later, the performance and efficiency of the method can be evaluated well based on both *ACC* and *TPR* measures.

It is seen that for most folds, after nearly six runs, the mineral database **h_db** becomes stable, with most mineral samples being included in the database. It is interesting to note that some specific samples in most folds, as shown in Figure 8, are wrongly included or are not appended to the database. For example, in all folds except fold 4, the same quartz mineral sample is wrongly included in the orthoclase class of the database (because in fold 4, the sample is introduced in the quartz class of the initial database).

As shown by the diagrams, almost all Pl, Or, and Mc samples are appended to the **h_db**. Additionally, in some cases, a few samples are falsely added to the database in the initial runs of the updating procedure; e.g., Pl in fold 6, or Mc in fold 9; however, subsequent runs of the updating algorithm have improved the results, since, after each update, the

acceptance circles change, which accordingly modifies the new database; i.e., some samples might be included in or removed from the database.

As a further experiment, we introduced all 200 samples as the initial database to investigate the response of the CARUS updating system. The wrongly located samples as well as those which are not included (i.e., removed or substituted) in the initial database are shown in Figure 11. It is interesting that these mineral samples are mostly observed in the results of the ten folds introduced in Figure 8. The wrongly located minerals are clearly different from typical templates of their associated mineral classes and their confusion is thus naturally expected.

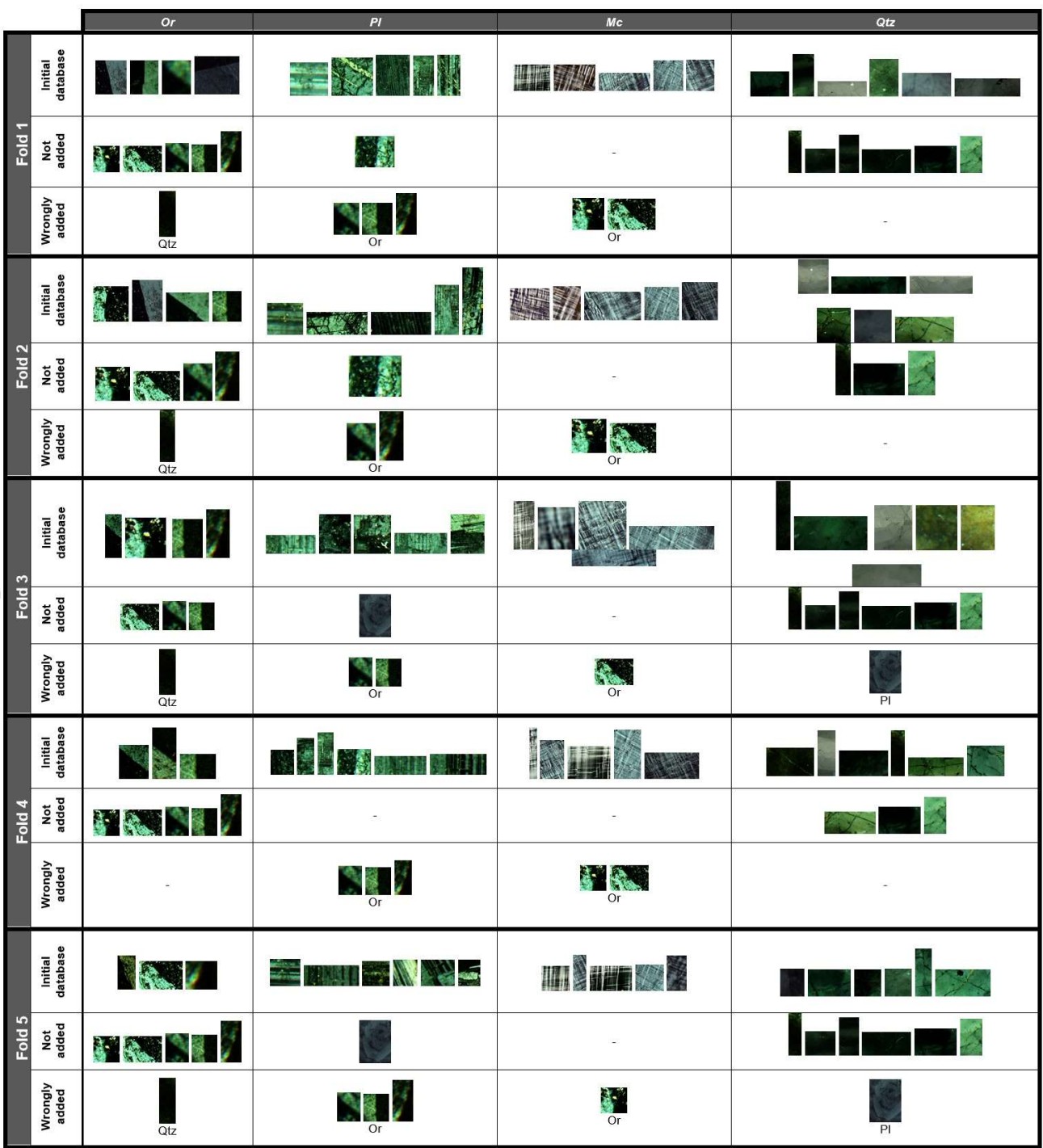

**Figure 9.** *Cont.*

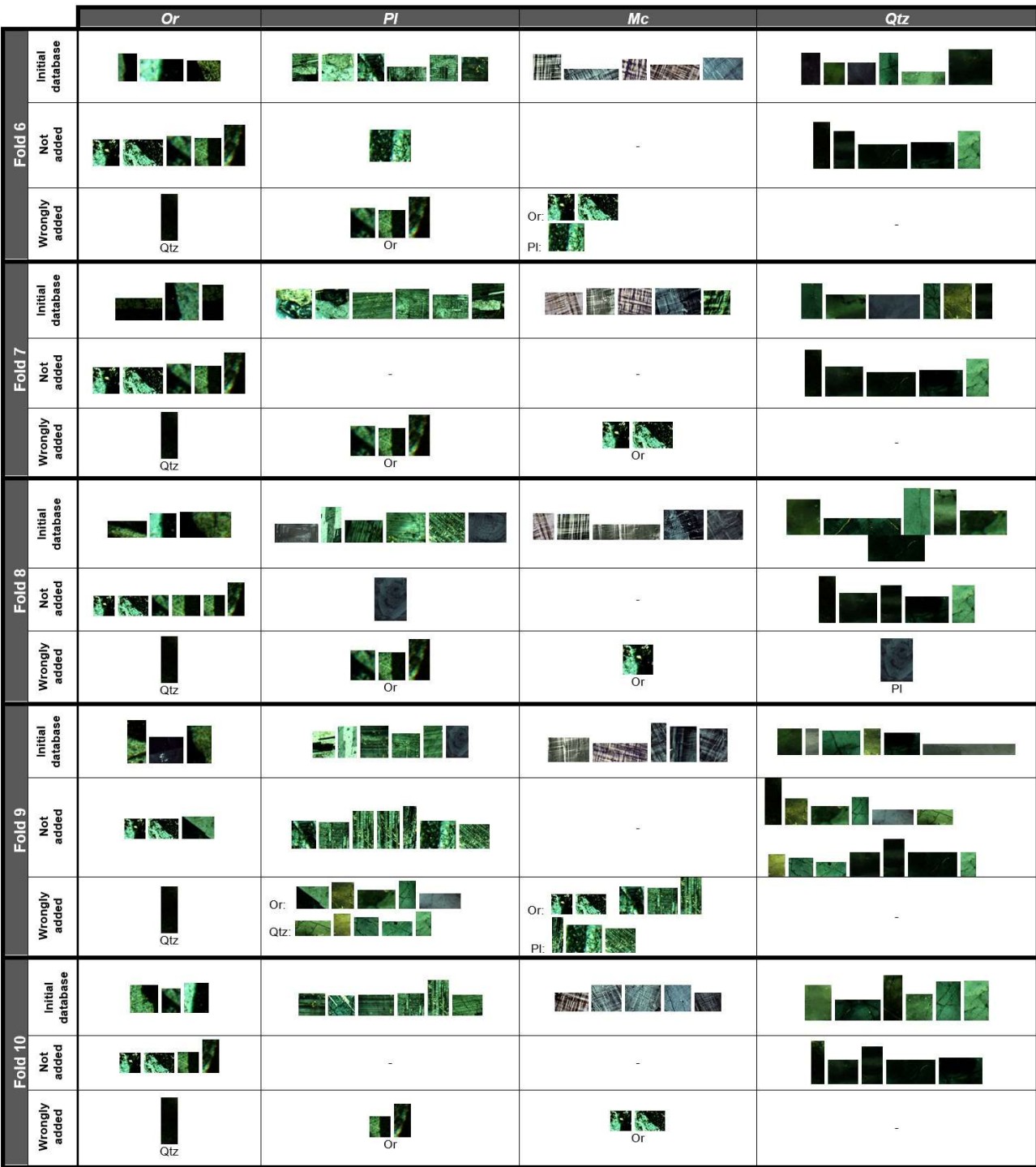

**Figure 9.** Initial database, wrongly added minerals, and minerals which are not included in their associated classes, obtained for all 10 folds.

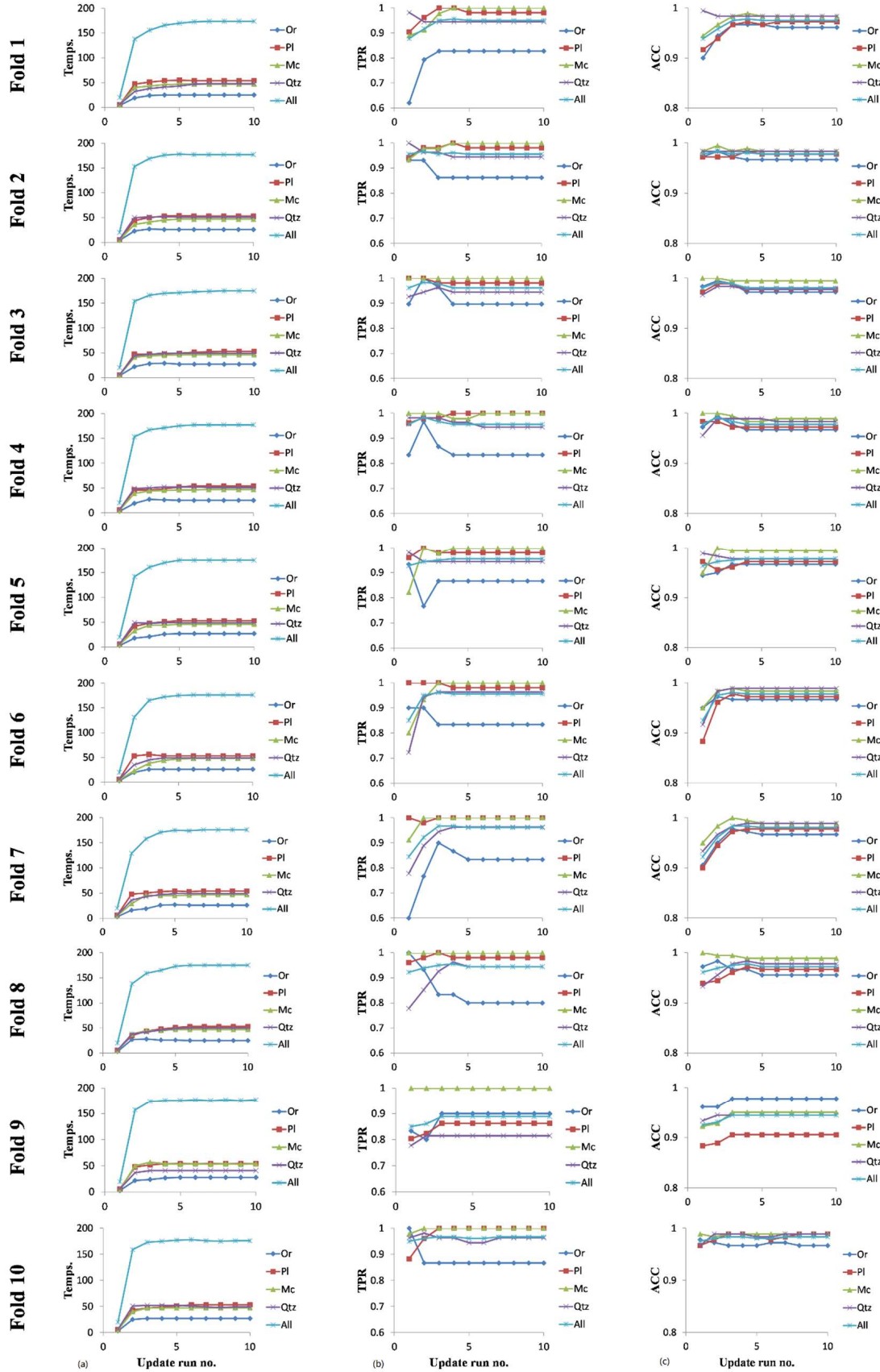

**Figure 10.** Convergence diagrams for 10 CARUS algorithm runs obtained for each fold; (**a**) *h_db* size; (**b**) *TPR*; (**c**) *ACC*.

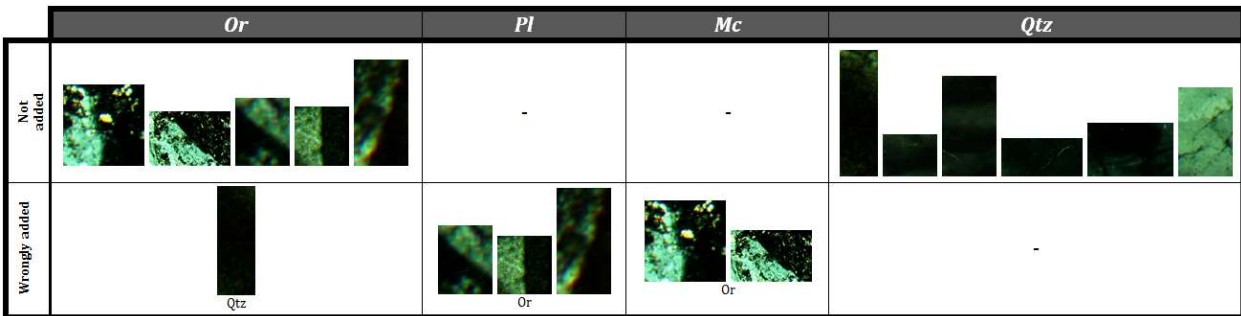

**Figure 11.** Wrongly added minerals, and minerals which are not included in their associated classes, for the case of all 200 samples as initial database.

*5.4. Comparison of CARUS Classification Method with kNN*

Different schemes have been used for the classification task, among which the k-nearest neighbors algorithm has obtained much reputation and application. In kNN classification, the class membership of an object is reported as the output, on the basis of a majority vote of its neighbors in the feature space; i.e., the object is assigned to the class most common among its k nearest neighbors, where k is a positive, typically small integer. However, the mineral classification method used in this research is based on a comparison of mean distances of the object (unknown mineral) in the feature space (LBP histogram), as introduced in Section 4. In this experiment, the two classification methods are compared. Again, the 10 mineral dataset folds, each with 20 sample images, are used, from which one fold is considered as the training set and the other fold as the testing set. kNN with Euclidean distance and three neighboring samples (which has been proved to be the most efficient option) is employed. The results, along with those of the proposed method, are introduced in Table 2. The reported discriminatory power values are the mean values of the 10 folds. It is clearly seen that the proposed method has efficiently performed better than kNN.

**Table 2.** The discriminatory power results for the CARUS classification method and kNN riLBP (16,2) is used as the texture operator.

| Method | Discriminatory Power |
|---|---|
| kNN | 0.825 |
| CARUS (before updating) TPR | 0.91 |
| CARUS (before updating) ACC | 0.955 |
| CARUS (after updating) TPR | 0.95 |
| CARUS (after updating) ACC | 0.975 |

## 6. Limitations and Future Work

The focus of this study is on the classification and identification of four main rock-forming minerals showing typical textural features. For the task of rock classification, however, identification of all rock-forming minerals is essential. For the identification of other minerals by means of automated digital microscopy, color features are proven to be efficient [17]. In future studies, combining color-based mineral identification with CARUS might be applied for the task of rock classification. Moreover, it might be useful to apply CARUS for identifying other minerals based on their textural features as well. Lastly, an online database might be developed for enriching the mineral data and empowering CARUS introduced here by adding images taken by different microscopes.

## 7. Conclusions

In this study, texture identification and classification of minerals according to the images taken under crossed polarized light (XPL) is considered. Texture features are effective for automated identification of minerals with low birefringence, including quartz, plagioclase, and K-feldspar. It would be a difficult task to recognize these minerals unless texture features such as twinning or undulose extinction are considered. For the purpose of automated rock classification, the identification of these minerals is of significant importance. It is also notable that other minerals also show significant texture characterization and texture analysis is an important task for any MI scheme.

The three following steps were conducted to develop an efficient well-performing automatic mineral texture identification algorithm, CARUS:

1. **Texture feature selection:** For mineral texture identification, noise and rotation are two main problems, and contrast has a key role in better recognition; thus, an LBP operator benefits from contrast and reduces the negative effects of noise and rotation is of much interest. The rotation-invariant local binary patterns, equipped with a complementary contrast measure, are proved to be a well-performing option for this purpose. It was shown that the selected rotation-invariant LBP can effectively outperform other texture features, such as GLCM, in the MI problem.

2. **Classification algorithm:** Mineral textures might be affected by structural and non-structural conditions; so, for reliable and robust TMI automation, a reasonable flexible classification algorithm is required. Here, in this study, a new classification scheme was designed based on mean distances between the LBP histogram of the unknown mineral with those belonging to a mineral class. The TMI results showed accuracies above 95% and proved to be better than the well-known kNN classification scheme, especially when an initial well-adjusted mineral database is provided.

3. **Database updating procedure:** Since no standard mineral texture database is available in the mineralogy community, an updating procedure was designed to improve any local initial database. The updating procedure incorporates newly identified minerals in the database if they satisfy certain criteria. Since during the TMI procedure, some minerals might falsely be recognized and located in the database, iterative runs of the updating algorithm will stabilize the LBP histogram database. Thus, the proposed updating procedure can automatically extend initial databases and produce efficient and rich mineral texture databases. It is notable that the update algorithm automatically removed samples that have high differences from samples in a certain class.

The CARUS LBP-based texture identification algorithm can efficiently be integrated with color-based MI schemes for the task of rock classification in subsequent research.

**Author Contributions:** Conceptualization, S.A. and R.K.; methodology, S.A. and R.K.; software, S.A.; validation, S.A. and R.K.; formal analysis, S.A. and R.K.; investigation, S.A. and R.K.; data curation, S.A. and R.K.; writing—original draft preparation, S.A., R.K., M.K. and D.J.A.; writing—review and editing, M.K. and D.J.A.; visualization, S.A. and R.K.; supervision, R.K., M.K. and D.J.A. All authors have read and agreed to the published version of the manuscript.

**Funding:** This study received no external funding.

**Institutional Review Board Statement:** Not applicable.

**Informed Consent Statement:** Not applicable.

**Data Availability Statement:** Data can be accessed by contacting the corresponding author.

**Conflicts of Interest:** The authors declare no conflict of interest.

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
