# Peer review of "Mineral Texture Identification Using Local Binary Patterns Equipped with a Classification and Recognition Updating System (CARUS)"

_sustainability, doi:10.3390/su141811291_

Round 1

Reviewer 1 Report

The authors have reported the study on mineral texture identification using local binary patterns equipped with a classification and recognition updating System.  It is well written. In general, the main conclusions presented in the paper are well supported by the figures and supporting text. However, to meet the journal quality standards, the following comments need to be addressed

1. The abstract can be polished and improved. The novelty problem statement described by the authors should be emphasized to attract general readers by providing more insights on the experiments and corresponding observations. Also, the authors should elaborate the general applicability of the current work.

2. The introduction writing part need to be improved. Also, the writing and presentation of the introduction lacks a bit in clarity. The paper requires some amount of rewriting to clarify all aspects of it, especially the novelty and new findings of this work that need to be clearly mentioned.

3. Typographical errors: There are some minor grammatical errors and incorrect sentence structures. Please run this through a spell checker

4. Following references can be added as relevant deep learning references   ( see: Scientific Reports 11, 1447 (2021)  https://doi.org/10.1038/s41598-021-81216-5’  diease detection: Neural, Comput & Applic (2022). 

Reviewer 2 Report

What is the application of the modeled texture in industrial scale and multiscale design of problems.

The innovation presented in this paper should clearly be stated. What is new compared to what has already been worked on this mineral twinning by texture inspection? 

Improve the discussion section.

The five components of Table 1 requires to be differentiated with (a), (b), (c), (d) and (f) for clarity and its discussions maintained with these numbers. Figures 8 and 10 is difficult to read and to understand. Authors should presnted clearer and more eligible figures.

What code in 5.4 was used in the kNN algorithm? If it is the udual k-spiral code from python, then the authors are kindly requested to consider more recent codes and modify for purposes of innovation 

Authors should also improve the literature references. A work as important as this paper should present references richer 60. There are more recents works that are relevant in this area of research that could improve the literature strength of this paper. Authors should refernce more recent publications. 

The methodology should be improved for replication research purposes. Elaborate the methods applied so that future research efforts can rely on the detailed methodology of this present work to open new horizons of related researches
